# Skeletal Fracture Detection with Deep Learning: A Comprehensive Review

**DOI:** 10.3390/diagnostics13203245

**Published:** 2023-10-18

**Authors:** Zhihao Su, Afzan Adam, Mohammad Faidzul Nasrudin, Masri Ayob, Gauthamen Punganan

**Affiliations:** 1Center for Artificial Intelligence Technology, Faculty of Information Science and Technology, Universiti Kebangsaan Malaysia, Bangi 43600, Selangor, Malaysia; p115435@siswa.ukm.edu.my (Z.S.); mfn@ukm.edu.my (M.F.N.); masri@ukm.edu.my (M.A.); 2Department of Orthopedics and Traumatology, Hospital Raja Permaisuri Bainun, Ipoh 30450, Perak, Malaysia; gauthamenpd@gmail.com

**Keywords:** deep learning algorithms, bone fracture detection, X-ray images, SDG4

## Abstract

Deep learning models have shown great promise in diagnosing skeletal fractures from X-ray images. However, challenges remain that hinder progress in this field. Firstly, a lack of clear definitions for recognition, classification, detection, and localization tasks hampers the consistent development and comparison of methodologies. The existing reviews often lack technical depth or have limited scope. Additionally, the absence of explainable facilities undermines the clinical application and expert confidence in results. To address these issues, this comprehensive review analyzes and evaluates 40 out of 337 recent papers identified in prestigious databases, including WOS, Scopus, and EI. The objectives of this review are threefold. Firstly, precise definitions are established for the bone fracture recognition, classification, detection, and localization tasks within deep learning. Secondly, each study is summarized based on key aspects such as the bones involved, research objectives, dataset sizes, methods employed, results obtained, and concluding remarks. This process distills the diverse approaches into a generalized processing framework or workflow. Moreover, this review identifies the crucial areas for future research in deep learning models for bone fracture diagnosis. These include enhancing the network interpretability, integrating multimodal clinical information, providing therapeutic schedule recommendations, and developing advanced visualization methods for clinical application. By addressing these challenges, deep learning models can be made more intelligent and specialized in this domain. In conclusion, this review fills the gap in precise task definitions within deep learning for bone fracture diagnosis and provides a comprehensive analysis of the recent research. The findings serve as a foundation for future advancements, enabling improved interpretability, multimodal integration, clinical decision support, and advanced visualization techniques.

## 1. Introduction

In recent years, medical image processing based on machine learning algorithms has gained more attention, especially deep learning algorithms [1,2]. Compared with traditional methods, deep learning algorithms have the strength to extract features automatically [3,4]. Deep learning algorithms are always applied in X-rays and CT image processing [5,6], such as assessing the mineral bone density (BMD), detecting bone fractures [7], and recommending treatment [8]. In clinical practice, it is a tiring and time-consuming job for doctors to mark fracture parts manually, so deep learning applied in the computer vision field has inspired many scholars to solve issues in the medical image field.

In the past, many bone fracture detection studies [9,10,11,12] utilized manual feature extractors to generate feature vectors, including color features, texture features, and shape features. Afterward, more and more related research [13,14,15,16,17] was inclined to cooperate with machine learning classifiers to recognize fractures, which improved bone fracture image processing technologies. However, traditional manual feature extraction algorithms and machine learning classifiers use complex mathematical operations but have limited performance [18,19,20]. So, deep learning algorithms have become increasingly popular in recent years making traditional methods obsolescent gradually.

Selecting the appropriate references is important. They should be closely related to the main topic and written by known experts. In fast-changing areas like tech or medicine, it is good to choose newer materials. Also, opt for sources that many people have reviewed and approved. It is a plus if a lot of researchers have cited them in their work. Make sure the references are of good quality and that the ways in which they obtained their information are solid. We identified 337 records from database searches and other sources. After screening 267 of these records, 198 were excluded, leaving 57 full-text articles for assessment. Ultimately, 40 articles were chosen for analysis. This selection process is illustrated in Figure 1, using a PRISMA flow diagram to show the review process and exclusion of papers.

The hot directions in bone fracture diagnosis not only include the application of deep learning algorithms for different specific tasks but also include the evolvement of deep learning algorithms, from traditional bounding boxes to semantically rich representations, emphasizing attention mechanisms and heat maps for comprehensive fracture visualization [19,20,21,22,23]. Although there are existing reviews on the topic, the precious understanding of the various domains of this rapidly evolving domain is missing. The review in [19] does not analyze the architecture and process of these AI models, while another review [21] only selected eleven studies, which cannot cover the entire research field. Additionally, the review in [22] is too wide to provide precise guidance. Furthermore, while foundational knowledge is vital, its presentation without the correlation to performance metrics, as seen in the review in [23], explains much basic knowledge but is lacking in showing the key indicators of each AI model.

Addressing these problems, our motivation for this comprehensive review is due to a dire need to bridge the knowledge gaps, provide a comprehensive understanding of the current methodologies, and find a path for future advancements. By analyzing the accomplishments and problems of the existing approaches, we aim to provide researchers, developers, and clinicians with a clear roadmap.

## 2. Background

### 2.1. Task Definition

Publications related to computer vision generally use recognition tasks to decide whether the object is the target or not, while the classification task is to classify the object into a specific category. The detection task is to find the target with the bounding box, while the localization task is to specify the location information of the target [24]. At present, most papers do not define the concept of bone fracture recognition, classification, detection, and localization clearly, so the search results for specific tasks were mixed. Additionally, deep learning algorithms for computer vision have systemic models to address different tasks. However, the definition of the four tasks in many papers is inconsistent with the mainstream terminology rule.

Publications related to computer vision generally use recognition tasks to decide whether the object is the target or not, while the classification task is to classify the object into a specific category. The detection task aims to find the target with a bounding box, whereas the localization task specifies the location information of the target [24].

However, the current landscape faces several pain points. Most notably, there’s significant ambiguity in many research papers regarding the definitions of bone fracture recognition, classification, detection, and localization. This lack of standardized terminologies and concepts has led to confusion in research methodologies and outcomes, resulting in mixed search results for specific tasks. For instance, the paper by Yang T H et al. in 2017 was filled with images of fractures or non-fractures, but instead detected through classifying [25].

Also, while deep learning algorithms for computer vision have set definitions for different tasks, many articles do not follow this clear path. A big problem is that different studies use different terminologies for the same tasks, making them different from what most people use. This difference makes it hard to combine knowledge from different places and slows down the development of common approaches and answers in this area.

The dominant challenge, therefore, is developing a universally accepted definitional formula for these tasks, ensuring that research outputs are both consistent and comparable, quickening the progress in the field. If we define the formula of bone diagnosis task as Y=f(x), and here, x is the input image, and Y is the output diagnosis result. For the four different tasks, Y has different meanings. For the recognition task, Y∈{fracture or non−fracture}. For the classification task, Y∈{type A fracture,type B fracture,...,or non−fracture}. For the detection task, $Y=\{y_{1},y_{2},y_{3},...,y_{n}|y_{i}=(x,y,w,h)\},$ which is a collection of fracture bounding boxes. For the localization task, $Y={\left\{a_{i,j}\right\}}_{m\times n}$, which is the matrix of the probability of fracture at each location.

### 2.2. Basic Knowledge

Image processing algorithms usually have two main steps as illustrated in Figure 2. The first step is feature extraction and the second step is the specific modeling task, such as recognition, classification, detection, and localization. Feature extraction usually has two methods, i.e., manual feature extraction and deep learning algorithms. Bone fracture diagnosis research needs to draw lessons from the state-of-the-art in deep learning [22] because the application of deep learning algorithms always lags behind theory research. In deep learning algorithms, feature extraction is carried out on backbone architecture, which is the most basic part of deep learning algorithms [26]. Once the features have been extracted, only the specific modeling tasks will be performed, such as classification, detection, etc. Both machine learning classifiers and deep learning models play a key role in solving the specific problem after extracting features. The process of these four tasks will be reviewed in detail in this paper.

The recognition task is a binary classification task that identifies whether the X-ray bone images are fractured or not, based on feature extraction. The classification task in this review induces only multi-classification, such as transverse fractures, oblique fractures, spiral fractures, etc. [27]. On the other hand, the purposes of detection and localization are to find the fracture position, and they exploit different tools. The detection task makes use of bounding boxes and recognizes fractures within the bounding boxes, while localization marks the fracture position with the help of heat maps, key points, or key lines.

## 3. Methods

Among a large number of papers, this review selected 40 papers for analysis based on these criteria. All these papers focus on deep learning algorithms applied to X-ray bone images. Furthermore, they had to meet the requirements of being published within the past five years, showcasing state-of-the-art approaches, proving their effectiveness, and representing the field well. To summarize various research studies, this review separated these papers into four modeling tasks, which are recognition, classification, detection, and localization.

Some classical deep learning models applied to the four tasks were introduced. The bone part, aim, dataset size, methods, results, and conclusion are from each of the selected papers. The bone part means the part of the human bone these papers are working on, while the aim shows what the objectives are of these studies. The dataset size shows how many pictures are in the dataset and the proportion of the training set and test set. The method shows detailed deep learning models and processes of experiments. The results show the experimental results of the models while the conclusion shows the final influence of the models. The core information of the 40 papers including the published year, paper index, task, method, dataset size, and bone part are summarized in Table 1 for easier referencing, and the generic processing of the bone X-ray images is shown in Figure 2. Finally, some solutions addressing three issues and a discussion of the future research direction will be proposed.

The evaluation metrics used by these papers are mostly accuracy, precision, sensitivity (or recall), and AUC as below to measure the experimental results of the different models [15,16]:

1. Accuracy: The accuracy represents the proportion of true predictions from all the recognition/localization/classification/detection tasks. The accuracy values range from 0 to 1 to correctly predict the presence of fractures and non-fractures.
Accuracy=(TP+TN)/(TP+TN+FP+FN)

In predictive modeling, there are four key metrics to evaluate the performance of a classifier. True positives (TP) are the number of instances that are correctly predicted as positive. True negatives (TN) represent the instances that are correctly identified as negative. On the other hand, false positives (FP) denote the instances that are incorrectly predicted as positive. Lastly, false negatives (FN) are those instances that are incorrectly predicted as negative. These metrics are crucial in understanding the accuracy and precision of a model, especially in contexts where the cost of misclassification is high.

2. Precision: The precision represents the proportion of the true positives divided by all the correct recognition/localization/classification/detection tasks. The precision values range from 0 to 1 to report the ability of the model to identify fractures as a proportion of all positives.
Precision=TP/(TP+FP)

3. Sensitivity (or recall): The sensitivity represents the value of correct recognition/localization/classification/detection divided by all the positive samples. The sensitivity values range from 0 to 1 to report the ability of the model to correctly predict the presence of fractures.
Sensitivity=TP/(TP+FN)

4. Specificity: The specificity demonstrates the proportion of unbroken bones correctly recognized as non-fractured. The specificity values range from 0 to 1 to show the probability of having no fractures, conditioned on truly being non-fractures.
Specificity=TN/(TN+FP)

5. Area under the curve (AUC): The AUC is the area under the ROC curve. The AUC value can analyze full prediction scores without setting a threshold. The AUC value equals the area under the ROC curve to the *x*-axis, which exploits sensitivity as the ordinate and (1-specificity) as the abscissa.

6. Average precision (AP): The AP calculates the precision of the object detection model at different recall values. Given an object class, firstly, rank the detection results in descending order of confidence; secondly, calculate the precision and recall for each detection. Finally, compute the area under the precision–recall curve.

7. Mean average precision (mAP): The mAP provides a single score for the model’s performance across all classes. C is the number of object classes. $AP_{i}$ is the average precision for the ${ith}^{}$ object class.
$$mAP=\frac{1}{C}\sum_{i=1}^{C}AP_{i}$$

8. Pixel accuracy: The pixel accuracy is the percentage of pixels that are correctly classified. This metric is useful for assessing the overall localization capability for models.

## 4. Modeling Tasks

A clearer definition of the modeling tasks has been refined based on the content of the bone fracture diagnosis and mainstream terminology rule of the computer vision domain from the papers [24,25,28,29,30,31,32,33,34,35,36,37,38,39,40,41,42,43,44,45,46,47,48,49,50,51,52,53,54,55,56,57,58,59,60,61,62,63,64,65,66]. The modeling task includes recognition, classification, detection, and localization.

Recognition: the recognition task is to identify whether the X-ray image is fractured or non-fractured.Classification: the classification task is to not only recognize fractures and non-fractures but also classify the fracture types.Detection: the detection task is to find the fracture position and engage the suitable bounding boxes to surround bone fracture parts completely and identify them with the help of bounding boxes.Localization: the localization task is to localize the fracture position directly with key points, key lines, or heat maps instead of bounding boxes, according to the entire semantic information of the X-ray images.

Object detection algorithms usually have two parts; one is to find the region of interest, and the other is to classify regions of interest as true or false. The deep learning algorithms for object detection have two categories, i.e., two-stage detectors and one-stage detectors. Two-stage detectors produce candidate regions first and then determine if they are true or false, while one-stage detectors perform classification and localization directly without region proposals. These detectors must be based on the backbone to perform feature extraction, such as AlexNet, VGG-16, GoogleNet, etc. [67]. In the past, because the computer did not have enough storage and computing capability, researchers usually utilized manual feature extraction methods to extract the feature from images, and similarity metrics to classify the fractures directly [21]. With the progress in hardware technologies, machine learning classifiers were applied more in bone recognition and classification tasks [68,69]. Deep learning detectors are widely applied in bone fracture detection tasks with bounding boxes [22]. However, the bounding boxes ignore the semantic information about the bone fractures, such as in [49,52,57]. Therefore, some researchers tried to highlight the fracture position with a heat map based on the attention mechanism and Grad CAM [70,71]. In summary, the evolution of the computational techniques in bone fracture diagnosis has progressed from manual feature extraction to advanced deep learning detectors, with a current trend towards utilizing attention mechanisms and heat maps to offer a more semantically rich representation of fractures, thereby addressing the limitations of traditional bounding boxes.

### 4.1. Recognition

#### 4.1.1. Full-Connected Neural Network (FCNN)

FCNN is a basic deep learning model, which contains an input layer, hidden layers, and an output layer. Each layer has one or more neurons, and its number depends on the dataset and the complexity of the tasks, such as a three-layer neural network with 1024 input neurons [25,30], and a four-layer neural network with 16 input neurons [34]. Full-connected neural networks can be exploited for classification and regression. The output value of the regression can be any number, while that of the classification must range from 0 to 1. So, we must determine the activation function, such as Sigmoid and Softmax. Bone fracture recognition is a binary classification; most studies prefer the Sigmoid activation function. If some studies try to engage the Softmax activation function, they must take the probability of fractures and non-fractures as two independent output values. After designing the full-connected neural networks, the model needs to be trained. Generally, full-connected neural networks are trained as back propagation neural networks. The original row images have too much redundant information, so researchers usually engage manual feature extraction methods to generate feature vectors, such as Haar wavelet transfer [25], SIFT [25,30], and ADPO [34]. In essence, full-connected neural networks serve as foundational deep learning models with varying layers and neurons, commonly used for bone fracture recognition via binary classification. The choice of activation functions and feature extraction techniques play pivotal roles in enhancing their performance and accuracy.

#### 4.1.2. Convolutional Neural Network (CNN)

CNN is a classical feedforward neural network, the most important component of which is the convolutional filter. Additionally, the CNN can also have full-connected layers and pooling layers. In the same way, convolutional neural networks are also trained as back propagation neural networks and work as fully connected neural networks. The convolutional layers of the CNN can extract features from raw input images, so they usually do not cooperate with manual feature extraction methods [28,29,31,33,35,36,37,39]. The CNN is diverse, so researchers designed various different backbone models, such as Inception V3 [31], Resnet [31], Xception [31], InceptionResNet v2 [33], DenseNet [33], and AlexNet [36]. Because each model has its own advantages, Refs. [31,33] even engage ensemble algorithms to combine multiple convolutional neural networks, which obtains excellent results. In addition, Ref. [36] exploits machine learning classifiers for fracture recognition based on the feature vectors from AlexNet, but it will not make the model an end-to-end network. Additionally, originally developed for natural language processing, the Transformer architecture has been repurposed for medical image analysis, especially in fracture recognition, by segmenting the images into patch sequences [72]. Based on its distinctive self-attention mechanism, the model emphasizes potential fracture regions, ensuring precise and efficient analysis while preserving the essential spatial relationships crucial for medical diagnosis [73,74,75,76]. In other fields of medical image processing, many excellent models have also been innovated which can also be applied to fracture recognition, including FABNet [77], DDANet [78], LPCANet [79], and some AF-SENet [80]. In summary, convolutional neural networks (CNNs) stand out for their innate ability to directly extract features from raw images, eliminating the need for manual feature extraction, and have given rise to a variety of innovative backbone models, some of which are even amalgamated to harness the strengths of multiple networks, leading to superior outcomes in tasks like fracture recognition.

### 4.2. Classification

#### 4.2.1. Convolutional Neural Network (CNN)

The CNN also works on the classification task, with a bit of tuning. The variations in CNNs applied for classification include DenseNet [43], Dense Dilated attentive network [46], and ResNet [41,47]. Their secret key technology is utilizing the convolutional filter. Therefore, the classification task has multiple output values and the Softmax activation function can normalize these output values; these models always select the Softmax activation function rather than Sigmoid [41,42,43,46,47]. Compared with the recognition task, bone fractures in the classification task have more categories. So, the classification task needs more high-quality features for classification. For example, Ref. [42] exploits another interactive pipeline to ask the user to re-orient the femur bones to obtain better features. Additionally, Kang added a scaled variant layer and a hybrid and progressive loss function to ResNet to better classify fractures in 2020 [47]. In conclusion, CNNs for classification often utilize the Softmax function due to varied outputs, and as bone fracture classification demands richer features, innovations like interactive pipelines and enhanced layers have been introduced for superior results.

#### 4.2.2. Generative Adversarial Network (GAN)

The GAN is an unsupervised learning method, which has a generative model and a discriminative model. The generative model can generate samples, and the discriminative model can select samples. Thus, the GAN can produce a large number of samples like the original data. So, the GAN can be used to perform data augmentation, which will then be used with the original data as inputs to a customized residual network [44]. In essence, the generative adversarial network (GAN) employs both generative and discriminative models to produce data samples akin to original data, enabling its application in data augmentation to complement and enhance datasets for further network training.

### 4.3. Detection

#### 4.3.1. Region Convolutional Neural Network (R-CNN)

The R-CNN is a classical deep learning detector, also based on the CNN proposed in 2014 by R. Girshick et al. [81]. The R-CNN produces region proposals first, and region proposals include 2000 object candidates. Almost all these object candidates are warped RoIs (region of interest). Then, the warped RoIs are resized to a fixed size to feed into the CNN. Next, the CNN network extracts a 4096-dimension feature vector for each proposal from the feature maps of these warped RoIs. Finally, the authors exploit the SVM classifier to obtain confidence scores and the bounding box regressor to predict four parameters, i.e., the center coordinates of the box along with its width and height. In the post-processing, non-maximum suppression (NMS) must be used to reduce the redundant bounding boxes. If the intersection over union (IoU) of two boxes is more than the threshold, that means the boxes are overlapped. Then, the box with a higher confidence score will be chosen. Although the R-CNN is a basic detector, researchers can improve the R-CNN detector by introducing or combining it with different techniques, like the cascade R-CNN with attention mechanisms and atrous convolution [58]. The is because atrous convolution can help extract a rich feature map and attention mechanisms can help focus on important features [70]. In summary, the R-CNN, a foundational deep learning detector, employs a series of processing steps from region proposal to non-maximum suppression for object detection, and its versatility allows for enhancements, such as integration with attention mechanisms and atrous convolution, optimizing feature extraction and refining detection outcomes.

#### 4.3.2. Faster-Region Convolutional Neural Network (Faster R-CNN)

The Faster R-CNN is an improved method based on the R-CNN, which also engages a region proposal network (RPN). The region proposal network (RPN) was introduced in 2015 [82]. The RPN has a classifier that classifies whether the detected region is an object or not, and the bounding box regressor can adjust the length and width of the bounding boxes. This makes the RPN faster than selective search algorithms. In the post-processing, non-maximum suppression (NMS) also must be used to reduce redundant bounding boxes. For the feature extractor, many classical convolutional neural network models can be the backbone of the Faster R-CNN, such as Inception-v4 [49] and Inception-ResNet [50]. In conclusion, the Faster R-CNN enhances the R-CNN by introducing the region proposal network (RPN) for quicker region proposals and maintains optimal bounding box selection through NMS, all while supporting various classical CNN backbones for superior feature extraction.

#### 4.3.3. Feature Pyramid Network (FPN)

The FPN is an upgraded method based on the Faster R-CNN. In 2017, T.-Y. Lin et al. combined the Faster R-CNN and FPN (feature pyramid network) [83]. The use of image pyramids to obtain feature pyramids (or featurized image pyramids) at multiple levels is a common method to improve the detection of small objects. In the post-processing, non-maximum suppression (NMS) also must be used to reduce redundant bounding boxes. The application of the FPN also needs to select a suitable backbone, such as ResNet-50 [54] and Res-Next-101 [57]. In essence, the FPN, an evolution of the Faster R-CNN, leverages feature pyramids to enhance small object detection, retains NMS for optimal bounding box selection, and pairs effectively with various backbones like ResNet-50 and Res-Next-101 for advanced feature extraction.

#### 4.3.4. You Only Look Once (YOLO)

YOLO is a series of one-stage detection algorithms, including YOLO v1 [84], YOLO v2 [85], etc. Two-stage detectors solve object detection at the classification stage following the proposal stage. However, in 2016, J. Redmon et al. proposed YOLO [84] (You Only Look Once), which reframed the detection task as a regression problem, directly predicting the image pixels as objects and their bounding box attributes. In the post-processing, non-maximum suppression (NMS) also must be used to reduce redundant bounding boxes. For example, Ref. [58] implements YOLO v5 applied in bone fracture detection to prove that two-stage detectors are much better than one-stage detectors in bone fracture detection. In summary, YOLO transforms object detection by viewing it as a regression problem, directly predicting bounding boxes. Despite its innovation, comparisons, like in [58], indicate that two-stage detectors might have an edge in bone fracture detection.

#### 4.3.5. RetinaNet

RetinaNet is a one-stage detection algorithm. In 2020, T. Lin and others proposed RetinaNet, which exploits the FPN to improve its performance [86]. Each layer from the FPN is passed to the subnets, enabling it to detect objects at various scales. The classification subnet predicts the object score for each location while the box regression subnet regresses the offset for each anchor to the ground truth. For example, Ref. [53] exploits RetinaNet to obtain a good performance in bone fracture detection, which is close to the two-stage detector’s performance. In conclusion, RetinaNet leverages the FPN and subnets to detect objects across scales, with distinct roles for classification and box regression, the performance of which in bone fracture detection approaches that of two-stage detectors.

### 4.4. Localization

#### 4.4.1. U-Net

The U-net model is a method for semantic segmentation. In 2015, RonNeberger O et al. proposed U-net for biomedical image segmentation, which is a U-shaped architecture with convolutional operation, up-sampling, and down-sampling process as well [87]. U-net can classify each pixel of the original input images as an object or background. So, U-net can localize the outline of bone fractures without bounding boxes. For example, Ref. [61] developed the extension of U-net, which can show the fracture probability and heat map. In conclusion, U-net, designed for biomedical segmentation, localizes bone fracture outlines without bounding boxes, with extensions visualizing fracture probabilities and heat maps.

#### 4.4.2. Fully Convolutional Network (FCN)

The FCN is used for semantic segmentation. In 2015, Long J et al. proposed the fully convolutional network, which uses transposed convolution layers instead of full-connected layers following common convolution layers [88]. The design can make the output image keep the same size as the input image. The pixels of the output image show different colors between objects and backgrounds. So, the FCN can localize the outline of the bone fractures without bounding boxes like the U-net model. For example, an FCN variant called MIL-FCN can obtain mined fractured localized regions of interest, whose backbone is DenseNet-121 [64]. In essence, the FCN innovates in semantic segmentation, maintaining consistent input-output image sizes for precise fracture localization without bounding boxes, as demonstrated by variants like MIL-FCN with a DenseNet-121 backbone.

#### 4.4.3. Spatial Transformer

The spatial transformer is a deep learning model that adopts self-attention mechanisms. In 2015, Jaderberg M et al. proposed the spatial transformer, which can distribute different weights to feature values according to their spatial importance [89]. For example, Ref. [62] adopts the spatial transformer to localize bone fractures. The spatial transformer can focus on the fracture features, so visualization of the spatial transformer can localize bone fractures. In conclusion, the transformer utilizes self-attention to assign weights based on spatial importance, allowing models to hone in on and visualize fracture features effectively.

#### 4.4.4. Gradient-Weighted Class Activation Mapping (Grad-CAM)

Gradient-weighted class activation mapping is a visualization method that can obtain a class activation map through the weighted sum of each feature map. In 2016, Selvaraju R R proposed Grad-CAM for the visualization of deep learning algorithms [90]. The Grad-CAM obtains each corresponding weight from the gradient of feature maps to visualize feature maps. For example, both [63,66] exploit gradient-weighted class activation mapping to localize bone fractures. However, they must recognize the bone fracture with CNN to obtain feature maps of bone fractures on the images before visualization. In summary, Grad-CAM visualizes deep learning features using weighted sums after initial detection with CNNs.

## 5. Discussion

The summary of the reviewed papers is shown in Table 1 and the visualized mind map according to its modeling tasks is in Figure 3. Deep learning techniques have revolutionized the diagnosis of bone fractures, focusing on various parts like the femur, wrist, and shoulder. Researchers employ these techniques across different diagnostic methods: recognition, which is a binary determination of fracture presence; classification, which discerns the specific type of fracture; detection, with most studies favoring the accuracy of two-stage detectors; and localization, which offers enhanced visualization through tools like heat maps. Each method showcases the versatility and potential of deep learning in improving the precision of bone fracture diagnosis.

Almost all the studies have their own focus on a certain bone part, such as the femur, wrist, shoulder, hip, hand, humerus, elbow, pelvis, ankle, etc. The pool of research on the different types of bones in the human body can be seen in the word cloud diagram in Figure 4. The most popular bone part included in this review paper is the femur, followed by the wrist, and the rest, following the font size in a decreasing manner. As different types of bone have different features, focusing on training and learning on one type of bone has a better performance for algorithms. However, researchers nowadays are inclined to work on finding a more generic model that can work on different parts simultaneously, such as the femur, wrist, shoulder, etc., which detect fractures in the shoulder, wrist, and ankle [29,36,57,61]. The most widely used public radiographic image dataset is MURA, which has 9045 healthy and 5818 unhealthy musculoskeletal X-ray images [29]. These images include the hand, humerus, finger, wrist, elbow, shoulder, and forearm. However, most papers collected datasets from hospitals, such as Chang Gung Memorial Hospital [63], Xi’an Honghui Hospital [58], Seoul National University Bundang Hospital [40], etc. The quantitative range of these datasets is usually from 100 to 100,000.

The recognition task can be seen as a binary classification of fractures or not fractures. At first, feature extraction methods with traditional methods or a CNN would be carried out. Then, machine learning classifiers will use these features to classify the fractures. Additionally, if the CNN or FCNN were used to perform binary classification, a suitable activation function must be selected, such as Sigmoid and Softmax. Refs. [25,30,34] exploit traditional feature extraction methods to attain feature maps first. Ref. [25] preprocess images with Haar wavelet theorems. Ref. [30] exploits Haar wavelet transform and SIFT operators to perform feature extraction. Ref. [34] extracts 13 line features with adaptive differential parameter optimized (ADPO) methods and then feeds them into a convolutional neural network. Refs. [28,29,31,33,35,36,37,39] engage convolutional neural networks to extract features. And then [28,29,39], engage the Sigmoid activation function to perform binary classification, i.e., fractures or not fractures. While [31,35,37] engage the Softmax activation function, which has two outputs, i.e., probabilities of fracture and non-fracture. Refs. [31,39] exploit the ensemble algorithm to combine different CNNs to obtain better experimental results.

The activation function makes the neural network an end-to-end trained method. Ref. [36] engages SVM to perform bone fracture recognition, but features are extracted with a pre-trained neural network. Ref. [36] has two independent steps, so they are trained separately. This study compares the performance of SVM, ELM, and RF and finds that SVM is the best classifier for the task. Therefore most of the studies use their own dataset, and the direct comparison of the experimental results is meaningless. However, we can still draw some conclusions from these results. In the public dataset MURA, the SVM classifier [36], obtains an accuracy of 78.63% while deep learning algorithms [34,36] obtain an AUC of 86.95%. In their self-collected datasets, most of these studies [25,30] have an accuracy of more than 90%. The experimental model shows that the proposed method has good feasibility and robustness, compared with the accuracy and standard deviation of previous studies.

In this review, the classification task classifies fracture types. For example, there are five different fracture types [41], based on the location of the bone, including normal, greater tuberosity fractures, surgical neck fractures, three-part fractures, and four-part fractures. So, the classification task is more complex than recognition. Most of the classification studies use convolutional neural networks to extract features because the convolutional operation is much better than the manual feature extraction method. In addition, the Softmax activation function outperforms the Sigmoid activation function in multi-classification tasks. In order to perform better, Ref. [42] re-orients the femur bones and moves the fracture line to the image center. Ref. [44] expands the training set with the generative adversarial network (GAN). Ref. [46] designed the DDA layer (dense dilated attention module) to give scope to the advantages of attention mechanisms. In classification tasks, most of the studies [41,44,46,47] utilized state-of-the-art deep learning algorithms, which have a high accuracy, sensitivity, and specificity of near or more than 90% and have a high AUC of more than 0.9. These experimental results show that deep learning algorithms can improve performance and accelerate application in clinical practice.

Detection algorithms mainly have two-stage detectors and one-stage detectors. Most papers choose two-stage detectors to be the baseline algorithm because they are more accurate than one-stage detectors. Two-stage means the generation of regions of interest and the identification of regions of interest are performed in two stages. One-stage means the generation of regions of interest and the identification of regions of interest are carried out simultaneously in one stage. Only [58] implements YOLO to be a comparison to prove their attention-based cascade R-CNN model is better than YOLO because the attention module can help R-CNN focus on more important features. Except for [53], which exploits RetinaNet to detect bone fractures and classify them into different types, nearly all papers prefer the most popular two-stage algorithm Faster R-CNN to be the baseline algorithm, and paired with different backbones, including Inception v4, Inception-ResNet v2, VGG16, and ResNet-50. Additionally, some researchers have tried to improve the Faster R-CNN to different degrees. Ref. [56] designed a guided anchoring method (GA), while [60] adds a feature pyramid network to Faster R-CNN. Ref. [57] even designed the feature ambiguity mitigate operator to combine with different models. Ref. [59] engages a triplet attention mechanism to obtain richer hairline fracture features. In the detection task, setting the threshold of IoU is necessary and researchers usually assign it to 0.5 or higher. Most of these models [49,50,51,54] are based on Faster R-CNN have a high mAP of more than 70%, better than YOLOv5 [58] with an mAP of about 60%. These studies show that the power of deep learning techniques can provide fast and accurate solutions to medical image analysis.

Localization has better visual results than the detection of bone fractures. Some papers use a heat map for the local fracture part, which is a visual way to mark fractures. A heat map is a probability map, which means it displays the probability of fracture in the corresponding point. Ref. [61] proposed an improved U-Net architecture to produce the probability of each place in the image directly. Ref. [62] generates visual heat maps based on the attention model (USTN) and self-transfer learning (STL). Ref. [64] produces a colorful probability map to mark the fracture position. Refs. [63,66] perform classification first and generate heat maps with the Grad-CAM method. In localization tasks, these studies [61,64] still need to set a threshold of IoU, and then obtain good accuracy, sensitivity, specificity, and an ROC of more than 90%. These studies prove that bone fracture marking has better ways, such as heat maps, key points, and key lines, instead of bounding boxes.

This review points out the strengths and weaknesses of various deep learning models applied in bone fracture diagnosis. A profound understanding of deep learning models can help researchers make more improvements in the application of bone fracture diagnosis, as theoretical research of algorithms is advancing faster than the application. In addition, multimodal information to assist X-ray image diagnoses is starting to attract more attention. For example, Refs. [32,40,45] use not only X-ray images but also patient demographic information and clinical data in their research. Another example of multimodal diagnosis was published by P. Tobler et al. in 2021, in which distal radius fractures were detected and classified with the help of radiology reports [38].

## 6. Solutions following the Three Issues

This review makes corresponding solutions to our proposed three issues and suggests future development.

First, with regard to the inconsistency between the definition of modeling tasks (i.e., recognition, classification, detection, and localization) in some papers with regard to bone fracture diagnosis and mainstream computer vision domain, we have refined a clearer definition [25,28,29,30,31,32,33,34,35,36,37,38,39,40,41,42,43,44,45,46,47,48,49,50,51,52,53,54,55,56,57,58,59,60,61,62,63,64,65,66]. This review formulates the standardized terminology for the diagnosis of bone fractures using X-ray images to link up with the theoretical research of deep learning algorithms. Because theoretical research of deep learning algorithms for computer vision is always ahead of its application [22], research about bone fracture diagnosis must absorb the latest research results from the corresponding state-of-the-art deep learning algorithms. Keeping consistent with mainstream terminology is efficient in drawing lessons from other advanced academic achievements.

Second, this review concludes a general processing workflow to fill the research gap in which existing reviews do not analyze technical details [19]. Our proposed general processing workflow can help researchers learn the structure and process of various deep learning algorithms applied in bone fracture diagnosis. Additionally, other scholars can also find some research gaps in the framework to perform some novel studies, which is shown in Figure 3.

Third, addressing the deficiencies in the field of clinical application for deep learning algorithms, this review proposes three suggestions. Foremost, we should try to add the interpretability of deep learning algorithms to provide doctors with sound and logical principles or the mathematics behind the algorithm’s judgment [4,91]. Additionally, we should try to process multi-modal clinical information including X-ray images, CT images, patient process data in hospitals, and some baseline clinical information (age, sex, body mass index, glucocorticoid use, and secondary osteoporosis) for personalized treatment and therapeutic schedule recommendation [92,93,94,95,96,97].

In the domain of future research directions, several important areas emerge. One significant area is algorithmic interpretability. With the rise of emerging techniques in explainable AI, there’s potential to make decision-making processes in deep learning algorithms for skeletal diagnosis more transparent [90]. Another crucial direction is the development of algorithms that can handle multimodal data combining image data and text data for clinicians [97]. Given the vast and diverse range of clinical data available, it is urgent to create algorithms that can efficiently integrate and process information, thereby offering personalized diagnostic and treatment recommendations. Additionally, as the domain of visualization techniques continues to advance, there’s a growing need to incorporate methodologies that can aid clinicians in better understanding and interpreting results from deep learning models [98]. A notable example includes the potential of achieving more accurate and clinically relevant 3D modeling derived from 2D X-ray images [99].

## 7. Conclusions

As medical AI systems are applied in clinics gradually, deep learning algorithms have greater research value. This review expounds on various related approaches in X-ray bone images based on deep learning algorithms. These studies engage several kinds of machine learning algorithms and obtain excellent results in different public datasets or self-collected datasets. This review makes several valuable contributions, including summarizing the latest research findings to help us know the newest research focus and cope with the too-fast upgrading speed of AI models, defining the four bone diagnosis tasks clearly, summarizing a general processing workflow to handle these tasks, and pointing out valuable research directions for deep learning models applied in bone fracture diagnosis for other researchers. Despite deep learning algorithms demonstrating comparable performance to clinicians, their clinical application still faces the challenge of proving their trustworthiness. In the future, we will try to increase the interpretability of networks, process multimodal clinical information, provide therapeutic schedule recommendations, and develop advanced visualization methods to improve the clinical application of deep learning algorithms.

## Figures and Tables

**Figure 1 diagnostics-13-03245-f001:**
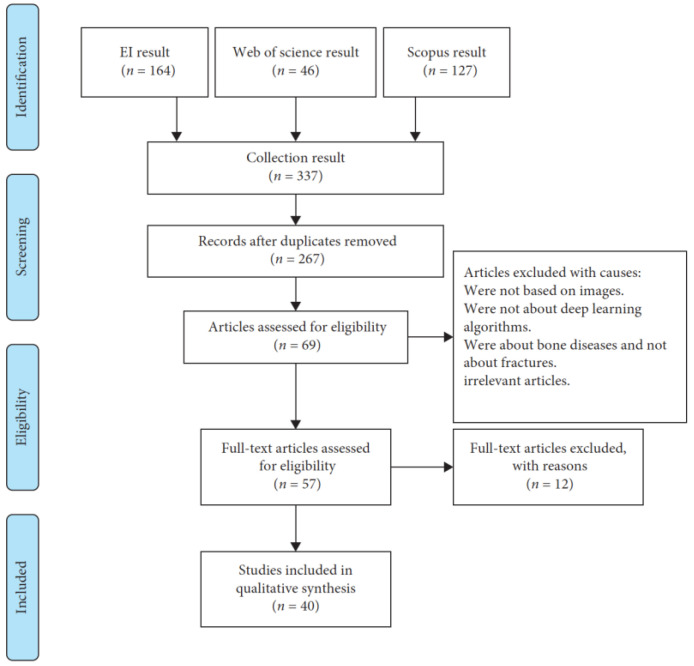
PRISMA flow diagram of the review process and exclusion of papers.

**Figure 2 diagnostics-13-03245-f002:**
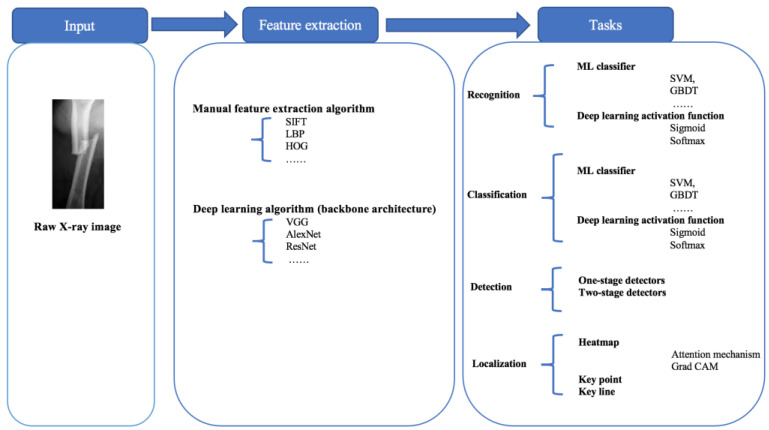
The processing graph of the four modeling tasks.

**Figure 3 diagnostics-13-03245-f003:**
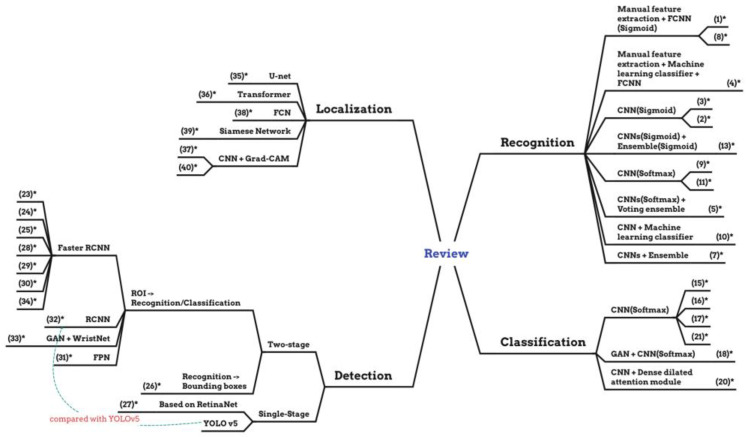
The mind map of the different approaches. (* means the number in Table 1).

**Figure 4 diagnostics-13-03245-f004:**
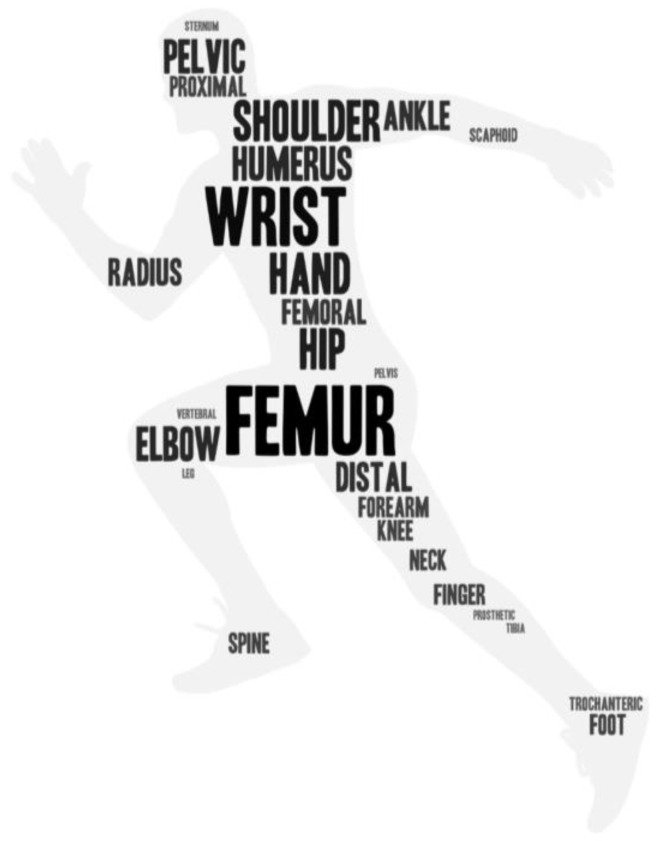
Frequency statistical chart of bone fractures parts in this review.

**Table 1 diagnostics-13-03245-t001:** Summary of the five features of the 40 approaches.

No	Year	Paper	Task	Method	Dataset Size	Part of Bone
(1)	2017	[25]	Recognition	Original images → Haar wavelet applied and reduced size image → BPNN (A three-layer neural network with 1024 input neurons, Sigmoid)	Training set: 30 images; testing set: 70 images	Null
(2)	2018	[28]	Recognition	Random forest regression voting constrained local model (RF-CLM) → registered patches → CNN (Sigmoid)	1010 pairs of wrist radiographs (PA, and LAT)	Wrist
(3)	2018	[29]	Recognition	View(s) → 169-layer convolutional neural network → arithmetic mean of output probabilities → probability of abnormality (Sigmoid)	MURA dataset	Shoulder, humerus, elbow, forearm, wrist, hand, and finger
(4)	2019	[30]	Recognition	Three steps: Haar wavelet transform and scale-invariant feature transform (SIFT) → k-means clustering based ‘bag of words’ methods → a classical back propagation neural network that contains 1024 neurons in 3 layers	Training set: 70 images; test set: 230 images	Null
(5)	2019	[31]	Recognition	A voting ensemble method (Inception V3, Resnet, and Xception convolutional neural networks → Softmax, two classes, normal, abnormal cases)	Validation and test sets: 240 views; training set: 1441 views	Ankle
(6)	2019	[32]	Recognition	Inception-v3 CNN architecture47	23,602 hip radiographs	Hip
(7)	2019	[33]	Recognition	Ensemble algorithm (InceptionResNetV2 CNN and a DenseNet CNN)	12,742 routine clinical VFA images	Vertebral
(8)	2019	[34]	Recognition	Input → (1) Standard line-based fracture detection; (2) adaptive differential parameter optimized (ADPO) line-based fracture detection → PCA → 13 line features → ANN (FCNN, BPNN) → fractures and non-fractures	Training set: 20 X-ray images; test set: 23 X-ray images	Leg
(9)	2020	[35]	Recognition	Deep CNN → Softmax (healthy, fracture)	4000 images	Null
(10)	2020	[36]	Recognition	AlexNet (extract features) → PCA → support vector machine (SVM), extreme learning machine (ELM) and random forest (RF)	MURA dataset	Shoulder, forearm, finger, humerus, elbow, hand and wrist
(11)	2020	[37]	Recognition	Convolutional neural network (CNN) architecture (five blocks, each containing a convolutional layer, batch normalization layer, rectified linear unit, and maximum pooling layer, Softmax, 2 classes, fractures, non-fractures)	234 frontal pelvic X-ray images	Femoral neck
(12)	2021	[38]	Recognition	ResNet18 DCNN	15,775 frontal and lateral radiographs	Distal radius
(13)	2021	[39]	Recognition	a new ensemble learning model (input → ResNet, ResNeXt, DenseNet, VGG, InceptionV3, and MobileNetV2 + Spinal FC layer → EL1, EL2, EL1, EL2 (spinal fully connected versions) models (Sigmoid)	The MURA dataset	Shoulder
(14)	2022	[40]	Recognition	A convolutional neural network (CNN)-based prediction algorithm called DeepSurv	Images and medical records from 7301 patients	Spine
(15)	2018	[41]	Classification	ResNet-152	1891 plain shoulder AP radiographs	Proximal humerus
(16)	2019	[42]	Classification	Convolutional neural network with two diagnostic pipelines	796 images: 97 AFF images and 399 NFF images	Femur
(17)	2019	[43]	Classification	DenseNet	750 images	Femur
(18)	2020	[44]	Classification	A customized residual network with Softmax classifier	1444 hip radiographs from 1195 patients	Femoral neck
(19)	2020	[45]	Classification	An encoder-decoder structured neural network	786 anterior–posterior pelvic X-ray images and 459 radiology reports acquired from 400 patients	Pelvic
(20)	2021	[46]	Classification	A novel dense dilated attentive (DDA) network	390 X-ray images	Femur trochanteric
(21)	2022	[47]	Classification	A scale-variant network (ResNet + a scaled variant (SV) layer + a hybrid and progressive (HP) loss function)	31/A1: 117 images; 31/A2: 125 images; 31/A3: 128 images	Femur trochanteric
(22)	2022	[48]	Classification + Detection + Localization	Vancouver Classification System (type A, B, C) (Densenet161, Resnet50, Inception, VGG and Faster R-CNN, RetinaNet)	1272 X-ray images	Periprosthetic femur
(23)	2019	[49]	Detection	Faster-R-CNN → Inception-v4	2340 AP wrist radiographs from 2340 patients	Distal radius
(24)	2019	[50]	Detection	Faster R-CNN (Inception-ResNet)	7356 wrist radiographic images	Wrist
(25)	2019	[51]	Detection	Faster R-CNN	4476 images with labels and bounding boxes for each augmented image	Distal radius
(26)	2020	[52]	Detection	Dilated Residual Network	16,019 unique radiographs	All
(27)	2020	[53]	Detection	RetinaNet object detection algorithm → ROIs (bounding boxes) → A densely connected convolutional neural network (DenseNet)	3034 hip images	Hip
(28)	2020	[54]	Detection	An anchor-based Faster R-CNN (ResNet-50 + pyramid networks (FPN))	2333 X-ray images	Femoral
(29)	2021	[55]	Detection	Faster region with convolutional neural network (Faster R-CNN) → CrackNet	3053 X-ray images	Null
(30)	2021	[56]	Detection	Faster R-CNN (GA module)	3067 X-ray radiographs	Hand
(31)	2021	[57]	Detection	FAMO model (ResNext- 101 + FPN)	1651 hand, 1302 wrist, 406 elbow, 696 shoulder, 1580 pelvic, 948 knee, 1180 ankle, and 1277 foot images	Hand, wrist, elbow, shoulder, pelvic, knee, ankle, and foot
(32)	2022	[58]	Detection	A deep convolutional neural network (cascade R-CNN + attention mechanism + atrous convolution)	1227 labeled X-ray images	Sternum
(33)	2022	[59]	Detection	Generative adversative network (GAN) → WrisNet is composed of two components (a feature extraction module + a detection module)	4346 X-rays	Hand
(34)	2022	[60]	Detection	Faster R-CNN network → ResNet	167 fractured samples and 194 normal samples for detection task, and 166 fractured samples and 194 normal samples for classification task	Scaphoid
(35)	2018	[61]	Localization	An extension of the U-Net architecture (two outputs: (1) fracture probability; (2) conditional heat map)	135,845 radiographs	Foot, elbow, shoulder, knee, spine, femur, ankle, humerus, pelvis, hip, and tibia
(36)	2018	[62]	Localization	Weakly supervised deep learning approach (spatial transformers (ST) + self-transfer learning (STL))	750 images from 672 patients taken	Proximal femur fractures
(37)	2019	[63]	Localization	DenseNet-121 → Grad-CAM	25,505 limb radiographs + 3605 PXRs	Hip
(38)	2019	[64]	Localization	MIL-FCN (DenseNet-121) → mined localized ROIs	4410 PXRs	Hip and pelvic
(39)	2020	[65]	Localization	A Siamese network (a spatial transformer layer)	2359 PXRs	Pelvic
(40)	2021	[66]	Localization	DCNN (EfficientNetB3) → Grad-CAM	8329 images	Scaphoid

## Data Availability

Not applicable.

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
