# Peer review of "Skeletal Fracture Detection with Deep Learning: A Comprehensive Review"

_diagnostics, 2023, doi:10.3390/diagnostics13203245_

Round 1

Reviewer 1 Report

1.   The author's motivation for the study was "Existing reviews...". This seems too general. Could the author please summarize a detailed motivation for the study in depth?

2.   The layout of this review manuscript does not make much good sense. A good review paper should be in the form of a rigorous definition of the problem being studied in the field. And it also summarizes the current state of research. Finally, it gives some directions with great research potential. After I read this review manuscript, I did not get to the summary and generalization of the research problem. Secondly, I also didn't see the authors suggesting a couple of specific hot directions for research. This is frustrating.

3.   In section 2.1, the analysis of the research task is not sufficient. The authors should add the pain points and difficult issues of the research task and its definitional formula. 

4.   In section 3, the description of the metrics is not comprehensive enough. Obviously, there are many more meaningful metrics for target detection, semantic segmentation and localization. The authors are asked to add them.

5.   In section 4.1, the description of computer vision methods is inadequate. Please add the SOTA model of CNN for medical image analysis.

[1].     "FABNet: fusion attention block and transfer learning for laryngeal cancer tumor grading in P63 IHC histopathology images," IEEE Journal of Biomedical and Health Informatics, vol. 26, no. 4, pp. 1696-1707, 2021.

[2].     "Interpretable laryngeal tumor grading of histopathological images via depth domain adaptive network with integration gradient CAM and priori experience-guided attention," Computers in Biology and Medicine, p. 106447, 2022.

[3].     "LPCANet: classification of laryngeal cancer histopathological images using a CNN with position attention and channel attention mechanisms," Interdisciplinary Sciences: Computational Life Sciences, vol. 13, no. 4, pp. 666-682, 2021.

[4].     "AF-SENet: Classification of cancer in cervical tissue pathological images based on fusing deep convolution features," Sensors, vol. 21, no. 1, p. 122, 2020.

6.   The transformer has become one of the mainstream methods in computer vision and medical image processing. Please add the following SOTA model in section 4.1.

[1].     "A ViT-AMC network with adaptive model fusion and multiobjective optimization for interpretable laryngeal tumor grading from histopathological images," IEEE Transactions on Medical Imaging, vol. 42, no. 1, pp. 15-28, 2022.

[2].     "Breast tumor grading network based on adaptive fusion and microscopic imaging," Opto-Electronic Engineering, vol. 50, no. 1, pp. 220158-1-220158-13, 2023.

[3].     "ASI-DBNet: an adaptive sparse interactive resnet-vision transformer dual-branch network for the grading of brain cancer histopathological images," Interdisciplinary Sciences: Computational Life Sciences, vol. 15, no. 1, pp. 15-31, 2023.

[4].     "The Swin-Transformer network based on focal loss is used to identify images of pathological subtypes of lung adenocarcinoma with high similarity and class imbalance," Journal of Cancer Research and Clinical Oncology, pp. 1-12, 2023.

7.   The few suggested directions given in section 6 are too general. More topical research directions should be given in relation to specific research questions and technical problems.

N/A

Author Response

Comments 1: The author's motivation for the study was "Existing reviews...". This seems too general. Could the author please summarize a detailed motivation for the study in depth?

Response 1: Thank you for pointing this out. We agree with this comment. Therefore, we have revised the motivation as below:

“Against these problems, our motivation for this comprehensive review is from a dire need: to bridge the knowledge gaps, provide a comprehensive understanding of current methodologies, and find a path for future advancements. By analyzing the accomplishments and problems of existing approaches, we aim to provide researchers, developers, and clinicians with a clear roadmap.”

Comments 2:  The layout of this review manuscript does not make much good sense. A good review paper should be in the form of a rigorous definition of the problem being studied in the field. It also summarizes the current state of research. Finally, it gives some directions with great research potential. After I read this review manuscript, I did not get to the summary and generalization of the research problem. Secondly, I also didn't see the authors suggesting a couple of specific hot directions for research. This is frustrating. 

Response 2: Thank you for pointing this out. We agree with this comment. Therefore, we have revised a couple of specific hot directions for research and summary and generalization of the research problem as below:

“The hot directions in bone fracture diagnosis not only include the application of deep learning algorithms for different specific tasks but also include the evolvement of deep learning algorithms, from traditional bounding boxes to semantically rich representations, emphasizing attention mechanisms and heat maps for comprehensive fracture visualization [19-23]. Although there are existing reviews on the topic, a precious understanding of the various domains of this rapidly evolving domain is missing. The review [19] does not analyze the architecture and process of these AI models, while another review [21] only selected eleven works, which cannot cover the entire research field. Additionally, the review [22] is too wide to provide precise guidance. Furthermore, while foundational knowledge is vital, its presentation without the correlation to performance metrics, as seen in the review [23], explains much basic knowledge but lacking in showing the key indicators of each AI model.  ”

Comments 3: In section 2.1, the analysis of the research task is not sufficient. The authors should add the pain points and difficult issues of the research task and its definitional formula. 

Response 3: Thank you for pointing this out. We agree with this comment. Therefore, we have added the pain points and difficult issues of the research task and its definitional formula below:

“Publications related to computer vision, generally, use recognition tasks to decide whether the object is the target or not, while the classification task is to classify the object into a specific category. The detection task is to find the target with the bounding box, while the localization task is to specify the location information of the target [24]. At present, most papers do not define the concept of bone fracture recognition, classification, detection, and localization clearly, so the search results for specific tasks were mixed up. For instance, the paper by Yang T H et al. in 2017 was filled with images of fractures or non-fractures, but instead detected by classifying [25]. Besides, deep learning algorithms for computer vision have systemic models to address different tasks. However, the definition of the four tasks in many papers is inconsistent with the mainstream terminology rule.

Publications related to computer vision generally use recognition tasks to decide whether the object is the target or not, while the classification task is to classify the object into a specific category. The detection task aims to find the target with a bounding box, whereas the localization task specifies the location information of the target [24].

However, the current landscape faces several pain points. Most notably, there's a significant ambiguity in many research papers regarding the definitions of bone fracture recognition, classification, detection, and localization. This lack of standardized terminologies and concepts has led to confusion in research methodologies and outcomes, resulting in mixed search results for specific tasks. For instance, the paper by Yang T H et al. in 2017 was filled with images of fractures or non-fractures, but instead detected by classifying [25].

Also, while deep learning algorithms for computer vision have set definitions for different tasks, many articles don't follow this clear path. A big problem is that different studies use different terminologies for the same tasks, making them different from what most people use. This difference makes it hard to combine knowledge from different places and slows down the development of common approaches and answers in this area.

The dominant challenge, therefore, is developing a universally accepted definitional formula for these tasks, ensuring that research outputs are both consistent and comparable, quickening the progress of the field. If we define the formula of bone diagnosis task is (formula in the paper), and here (formula in the paper)  is the input image,  is the output diagnosis result. For different four tasks, (formula in the paper) has different meanings. For the recognition task,(formula in the paper). For the classification task, (formula in the paper). For the detection task, (formula in the paper), which is a collection of fracture bounding boxes. For the localization task, (formula in the paper), which is (formula in the paper) is the matrix of the probability of fracture at each location.”

Comments 4:   In section 3, the description of the metrics is not comprehensive enough. Obviously, there are many more meaningful metrics for target detection, semantic segmentation and localization. The authors are asked to add them.

Response 4: Thank you for pointing this out. We agree with this comment. Therefore, we have added many more meaningful metrics for target detection, semantic segmentation and localization as below:"

  1. Average Precision (AP): AP calculates the precision of your object detection model at different recall values. Given an object class, Firstly, rank detection results in descending order of confidence; Secondly, calculate the precision and recall for each detection. Finally, compute the area under the precision-recall curve.  
  2. mean Average Precision (mAP): mAP provides a single score for the model's performance across all classes. is the number of object classes.  is the Average Precision for the  object class.  

  1. Pixel Accuracy: Pixel Accuracyis the percentage of pixels that are correctly classified. This metric is useful for assessing the overall localization capability for models."

Comments 5:    In section 4.1, the description of computer vision methods is inadequate. Please add the SOTA model of CNN for medical image analysis.

[1].     "FABNet: fusion attention block and transfer learning for laryngeal cancer tumor grading in P63 IHC histopathology images," IEEE Journal of Biomedical and Health Informatics, vol. 26, no. 4, pp. 1696-1707, 2021.

[2].     "Interpretable laryngeal tumor grading of histopathological images via depth domain adaptive network with integration gradient CAM and priori experience-guided attention," Computers in Biology and Medicine, p. 106447, 2022.

[3].     "LPCANet: classification of laryngeal cancer histopathological images using a CNN with position attention and channel attention mechanisms," Interdisciplinary Sciences: Computational Life Sciences, vol. 13, no. 4, pp. 666-682, 2021.

[4].     "AF-SENet: Classification of cancer in cervical tissue pathological images based on fusing deep convolution features," Sensors, vol. 21, no. 1, p. 122, 2020.

Response 5: Thank you for pointing this out. We agree with this comment. Therefore, we have cited the four citation [90-93].

“In other fields of medical image processing, many excellent models have also been innovated, which can also be applied to fracture recognition, including FABNet [90], DDANet [91], LPCANet [92] and some AF-SENet [93].”

Comments 6:    The transformer has become one of the mainstream methods in computer vision and medical image processing. Please add the following SOTA model in section 4.1.

[1].     "A ViT-AMC network with adaptive model fusion and multiobjective optimization for interpretable laryngeal tumor grading from histopathological images," IEEE Transactions on Medical Imaging, vol. 42, no. 1, pp. 15-28, 2022.

[2].     "Breast tumor grading network based on adaptive fusion and microscopic imaging," Opto-Electronic Engineering, vol. 50, no. 1, pp. 220158-1-220158-13, 2023.

[3].     "ASI-DBNet: an adaptive sparse interactive resnet-vision transformer dual-branch network for the grading of brain cancer histopathological images," Interdisciplinary Sciences: Computational Life Sciences, vol. 15, no. 1, pp. 15-31, 2023.

[4].     "The Swin-Transformer network based on focal loss is used to identify images of pathological subtypes of lung adenocarcinoma with high similarity and class imbalance," Journal of Cancer Research and Clinical Oncology, pp. 1-12, 2023.

Response 6: Thank you for pointing this out. We agree with this comment. Therefore, we have cited the four citations [94-98] as below:

“Besides, originally developed for natural language processing, the Transformer architecture has been repurposed for medical image analysis, especially in fracture recognition, by segmenting images into patch sequences [94]. Based on its distinctive self-attention mechanism, the model emphasizes potential fracture regions, ensuring precise and efficient analysis while preserving essential spatial relationships crucial for medical diagnosis [95-98].”

Comments 7:    The few suggested directions given in section 6 are too general. More topical research directions should be given in relation to specific research questions and technical problems.

Response 7: Thank you for pointing this out. We agree with this comment. Therefore, we have added specific research questions and technical problems as below:

“In the domain of future research directions, several important areas emerge. One significant area is algorithmic interpretability. With the rise of emerging techniques in explainable AI, there's potential to make decision-making processes in deep learning algorithms for skeletal diagnosis more transparent [81]. Another crucial direction is the development of algorithms that can handle multimodal data combining image data and text data for clinicians [88]. Given the vast and diverse range of clinical data available, it's urgent to create algorithms that can efficiently integrate and process information, thereby offering personalized diagnostic and treatment recommendations. Additionally, as the domain of visualization techniques continues to advance, there's a growing need to incorporate methodologies that can aid clinicians in better understanding and interpreting results from deep learning models [101]. A notable example includes the potential of achieving more accurate and clinically relevant 3D modeling derived from 2D X-ray images [89].”

Reviewer 2 Report

Regarding the manuscript entitled " Skeletal Fracture Detection with Deep Learning: A Comprehensive Review in which  The authors present a comprehensive review of 40 recent papers from prestigious databases, including WOS, Scopus, and EI, to analyze and evaluate the use of deep learning models in diagnosing skeletal fractures from X-ray images. The authors establish precise definitions for bone fracture recognition, classification, detection, and localization tasks within deep learning, and summarize each study based on key aspects such as bones involved, research objectives, dataset sizes, methods employed, results obtained, and concluding remarks. Finally, the authors identify crucial areas for future research in deep-learning models for bone fracture diagnosis, including enhancing network interpretability, integrating multimodal clinical information, providing therapeutic schedule recommendations, and developing advanced visualization methods for clinical application.

This manuscript is well organized. It has a high clinical value and has filled the existing gap in the field of review studies related to the diagnosis and fracture of the skeleton.

It seems that this manuscript has an acceptable methodology and with less revision it can be suggested for publication. In order for this manuscript to be of acceptable quality, it is better to pay attention to these points.

1: In the abstract, write how many studies you found in the first stage and finally reached forty.

2: For review studies, I suggest that you definitely use Prisma Protocol to determine how many studies you have reached in the first step to the final number of forty articles.

3: In the review studies, it should be determined what your inclusion and exclusion criteria were. That seems to be the case PRISMA

I reflect it. Therefore, I suggest you get inspiration from the following review manuscripts.

"Deep Learning in the Detection and Diagnosis of COVID-19 Using Radiology Modalities: A Systematic Review"

3: Some parts do not have references, such as sections "The evaluation metrics" You can use the following very relevant references for these sections."A fast and efficient CNN model for B-ALL diagnosis and its subtypes classification using peripheral blood smear images"

"A mobile application based on efficient lightweight CNN model for classification of B-ALL cancer from non-cancerous cells: a design and implementation study"

Due to the lack of a complete and comprehensive reference regarding deep learning in the field of its applications

I also suggest to use this reference for more richness of deep learning section. This reference adds value to your list of manuscripts.

"Deep learning: Applications, architectures, models, tools, and frameworks: A comprehensive survey"

  4: On the other hand, because your study is an overview. Therefore, I suggest that you make suggestions for the applications of deep learning in the field of skeletal fractures as a " Suggestion for Future reseach" . For this purpose, I suggest that you present it in the conclusion section to be a road map for future researchers.

Author Response

Comments 1:  In the abstract, write how many studies you found in the first stage and finally reached forty.

Response 1: Thank you for pointing this out. We agree with this comment. Therefore, we have revised “To address these issues, this comprehensive review analyzes and evaluates 40 recent papers from 837 papers in prestigious databases, including WOS, Scopus, and EI. ”

Comments 2: For review studies, I suggest that you definitely use Prisma Protocol to determine how many studies you have reached in the first step to the final number of forty articles.

Response 2: Thank you for pointing this out. We agree with this comment. Therefore, we have added “Figure 1. PRISMA flow diagram of the review process and exclusion of papers”.

Comments 3: In the review studies, it should be determined what your inclusion and exclusion criteria were. That seems to be the case PRISMA

I reflect it. Therefore, I suggest you get inspiration from the following review manuscripts.

"Deep Learning in the Detection and Diagnosis of COVID-19 Using Radiology Modalities: A Systematic Review"

Response 3: Thank you for pointing this out. We agree with this comment. Therefore, we have added “Selecting the appropriate references is important. They should be closely related to the main topic and written by known experts. In fast-changing areas like tech or medicine, it's good to choose newer materials. Also, go for sources that many people have reviewed and approved. It's a plus if a lot of researchers have cited them in their work. Make sure the references are of good quality and that the ways they got their information are solid. We identified 337 records from database searches and other sources. After screening 267 of these records, 198 were excluded, leaving 57 full-text articles for assessment. Ultimately, 40 articles were chosen for analysis. This selection process is illustrated in Figure 1, using a PRISMA flow diagram to show the review process and exclusion of papers.” Based on the paper “Ghaderzadeh M, Asadi F. Deep learning in the detection and diagnosis of COVID-19 using radiology modalities: a systematic review[J]. Journal of Healthcare engineering, 2021, 2021.”

Comments 3: Some parts do not have references, such as sections "The evaluation metrics" You can use the following very relevant references for these sections."A fast and efficient CNN model for B-ALL diagnosis and its subtypes classification using peripheral blood smear images"

"A mobile application based on efficient lightweight CNN model for classification of B-ALL cancer from non-cancerous cells: a design and implementation study"

Due to the lack of a complete and comprehensive reference regarding deep learning in the field of its applications

I also suggest to use this reference for more richness of deep learning section. This reference adds value to your list of manuscripts.

"Deep learning: Applications, architectures, models, tools, and frameworks: A comprehensive survey"

Response 4: Thank you for pointing this out. We agree with this comment. Therefore, we have added the three citations [9-101].

[1] Ghaderzadeh M, Aria M, Hosseini A, et al. A fast and efficient CNN model for B‐ALL diagnosis and its subtypes classification using peripheral blood smear images[J]. International Journal of Intelligent Systems, 2022, 37(8): 5113-5133.

[2] Hosseini A, Eshraghi M A, Taami T, et al. A mobile application based on efficient lightweight CNN model for classification of B-ALL cancer from non-cancerous cells: a design and implementation study[J]. Informatics in Medicine Unlocked, 2023, 39: 101244.

[3] Gheisari M, Ebrahimzadeh F, Rahimi M, et al. Deep learning: Applications, architectures, models, tools, and frameworks: A comprehensive survey[J]. CAAI Transactions on Intelligence Technology, 2023.

Comments  4: On the other hand, because your study is an overview. Therefore, I suggest that you make suggestions for the applications of deep learning in the field of skeletal fractures as a " Suggestion for Future reseach" . For this purpose, I suggest that you present it in the conclusion section to be a road map for future researchers.

Response 5: Thank you for pointing this out. We agree with this comment. Therefore, we have added in the conclusion. “In the future, we will try to increase the interpretability of networks, process multimodal clinical information, provide therapeutic schedule recommendations, and develop advanced visualization methods to improve the clinical application of deep learning algorithms.”

Round 2

Reviewer 1 Report

The author has addressed all my concerns very well.